# "I am alive; my baby is alive": Understanding reasons for satisfaction and dissatisfaction with maternal health care services in the context of user fee removal policy in Nigeria

**Anthony Idowu Ajayi** *

Population Dynamics and Reproductive Health and Right Unit, African Population and Health Research Center, APHRC Campus, Nairobi, Kenya

* ajayianthony@gmail.com

## Abstract

**Data Availability Statement:** All relevant data are within the manuscript and its Supporting Information files.

### Background

The main policy thrust in many sub-Saharan Africa countries' aim at addressing maternal mortality is the elimination of the user fee for maternal healthcare services. While several studies have documented the effect of the user fee removal policy on the use of maternal health care services, the experiences of women seeking care in facilities offering free obstetrics services, their level of satisfaction and reasons for satisfaction or dissatisfaction are poorly understood.

### Methods

This study adopted a mixed study design involving a population survey of 1227 women of reproductive age who gave birth in the last five years preceding the study (2011–2015), 68 in-depth interviews, and six focus group discussions. Simple descriptive statistics were performed on 407 women who benefitted from the user fee removal policy, while the qualitative data were analysed using thematic analysis.

### Results

The overall level of satisfaction with care received was remarkably high (97.1%), with birth outcomes being the central reason for their satisfaction. Participants were also satisfied with both the process aspect of care (which includes health workers' attitude and privacy) and the structural dimension of care (such as, the cleanliness of health care facilities and availability of and access to medicine). From the qualitative analysis, prolonged waiting-time, the limited scope of coverage, mistreatment, disrespect and abuse, inadequate infrastructure and bed space were the main reasons why a few women were dissatisfied with care under free maternal health care.

**Funding:** The author received no specific funding for this work.

**Competing interests:** The author has declared that no competing interests exist.

## Conclusion

The findings establish a high level of beneficiaries' satisfaction with care under free maternal health policy in Nigeria, raising the need for sustaining the policy in expanding access to maternal health services for the poor. Nevertheless, issues relating to prolonged waiting-time, the limited scope of coverage, mistreatment, disrespect and abuse, inadequate infra-structure and bed space require attention from policymakers.

## Introduction

Most maternal deaths are preventable with the use of quality obstetrics services; however, many women do not have access to these services in sub-Saharan Africa[1–4], and especially in Nigeria [5, 6]. To address the suboptimal utilisation of maternal health care services, many countries in sub-Saharan Africa have introduced user fee removal policy. Even though there is evidence that free maternal health care policy is associated with increased uptake of services [7–10], some studies have also shown that the sharp increase in the rate of utilisation following the introduction of the policy is only temporary [11–13].

The literature appears to paint a dire picture of the effect of user fee removal policy on qual-ity of care, with some authors arguing that facilities offering free services had difficulty responding to the influx of patients; thus, leading clients to turn to paid services [14–18]. Also, it appears that the introduction of user fee exemption policy further exacerbates the problem of inadequate resources to the extent that services become unavailable after a while [16, 19, 20]. Kruk et al. [21] affirm that women bypass health facilities offering free services in prefer-ence for better quality care. Given this context, investigating the experiences of women who benefit from free maternal health care services, assessing their level of satisfaction and explor-ing reasons for their satisfaction or dissatisfaction is worthwhile. Besides, women' satisfaction with maternal health care services is critical to the continued use of the services or willingness to return to the health facilities [22].

Indeed, satisfaction with maternal health care services is a complex subject, often studied, but poorly measured [23]. Satisfaction with care is a multidimensional concept. Bramadat et al. [23] define satisfaction as a "positive feeling" or "affective response" to an event. When studying women's satisfaction with maternal health care services, three dimensions of care are discussed, which are structural, processes and outcomes. The structural aspect of care, accord-ing to Srivastava et al. [24], includes good physical environment, cleanliness of wards, theatres and toilets, and adequate human and material resources, while the process dimension of care relates to interpersonal and emotional support received from midwives or doctors, privacy and promptness. The outcome dimension of care, however, connotes the health status of the mother and the baby.

Studies that examine the level of satisfaction with maternal health care services have reported a varying level of satisfaction, but in general, a high level of satisfaction is consistently reported even in settings where the quality of care provided is considered poor [22, 25–29]. Thus, it is unclear whether high satisfaction means high-quality care, or whether satisfaction with one dimension of care equates with the overall satisfaction with care received. In other words, could a woman who is not satisfied with the process dimension of care still report a high level of satisfaction?

Nigeria has a huge burden of maternal deaths, with about 19% of all mortalities globally [3, 4]. User fee removal policy was introduced in 2010 to reverse the scourge of maternal deaths

and improve the health of women in the country. The policy makes antenatal, delivery care, and postnatal care, including caesarean section free for all pregnant women. Several studies have reported the poor quality of maternal healthcare services in Nigeria [30–35], as such, it remains unclear how the introduction of free maternal healthcare would impact the quality of care or user satisfaction, which are important factors that could influence demand or future use of the services.

While several studies have documented the effect of user fee removal policy on maternal health care services' utilisation [9, 36–40], experiences of women seeking care in facilities offering free obstetrics services, the women's level of satisfaction and reasons for satisfaction or dissatisfaction are less understood. The main thrust of this study is to examine the experiences of beneficiaries of free maternal health care services, assess their level of satisfaction and reasons for satisfaction and dissatisfaction with care received. The focus of this paper aligns with current scholarly focus and interest in experiences of respectful, women-centred, maternal health care services and drivers of users' satisfaction with care. This is essential, especially in the context of free maternal health care implementation, which is characterised as disruptive to the health system functioning and exacerbates the problem of poor quality services in the sub-Saharan Africa context [11].

## Materials and methods

### Study settings

The data analysed in this study were retrieved from the Mancofree study, and the full details of the method have been published elsewhere [5]. This study was conducted in three states which were selected purposively from two of the six geopolitical zones in Nigeria. The choice of the two geopolitical zones is an obvious one because of the uniqueness of their free maternal health programmes. Nigeria has a federal system of government, and as such, maternal healthcare policies vary across regions and states. There is variation in the implementation of maternal healthcare policy among the states. Ondo and Ekiti states were selected in the Southwestern geopolitical zone, and Nasarawa state was selected in Northcentral. The full details of the programme implementation in the selected study areas have been described elsewhere[41]; however, a snapshot of the key differences in the implementation of the free maternal health care programme are highlighted in Table 1.

### Study design

This study adopted a convergent mixed-methods design[42], involving a population-based survey of 1227 women within the reproductive age group who gave birth in the past five years preceding the survey (2011–2015), 68 in-depth interviews (IDIs) and six focus group discussions (FGDs) that all took place concurrently. The rationale for adopting a mixed-methods design was to gather nuanced data that helps develop an in-depth understanding of reasons for satisfaction and dissatisfaction with the quality of care in the context of free maternal health care, which is impossible to achieve by adopting only a quantitative method. Data collection took place between May and September 2016. A purposive sampling technique was employed in selecting participants for the qualitative aspect of this study. Participants who met the inclusion criteria were approached face-to-face by the research team and they discussed the study objectives with them and later asked for their willingness and permission to be part of the study participants. We selected a diverse group of women, including young and middle-aged mothers, as well as those who had given birth in primary health centres, secondary and tertiary facilities to ensure all views were captured. The interviews were conducted in the homes of the

**Table 1.  Differences in the implementation of free maternal health programme by states.**

|  | Ondo State | Ekiti State | Nasarawa State |
|---|---|---|---|
| Programme name | "Abiye" a term that means safe motherhood in Yoruba language | "Itoju ofe fun alaboyun" meaning free healthcare for pregnant women | Free health care for pregnant women |
| Media coverage and popularity of the programme | Very high | Moderately high | Popular in the urban area but less popular in rural areas |
| Agenda setting | The programme began with a pilot study to understand the reasons for delays in seeking maternal health services. Handling out of telephone to pregnant women to contact nurses was piloted in two local government areas. The scale-up of the programme, however, did not involve the telephone intervention. | The programme began with a planning meeting with key stakeholders including policymakers from the ministry of health and hospital managers. | The programme began with a planning meeting with key stakeholders, including policymakers from the ministry of health and hospital managers. |
| Scope of coverage | The policy covers medications, prenatal care, delivery care, caesarean section, and postnatal care. | The policy covers medications, prenatal care, delivery care, caesarean section, and postnatal care. | The policy covers medications, prenatal care, delivery care, caesarean section, and postnatal care. |
| Other benefits | As part of the pilot programme, the feasibility of delivery kits containing baby oil and clothes handed, mobile phone for communication with health workers and tricycle as ambulance was tested but was not scaled up during the programme implementation. | "Good Mama package" delivery kits containing baby oil and clothes handed out to some pregnant women at the initial stage of the programme implementation but was stopped due to lack of fund to continue. | "Good Mama package" delivery kits containing baby oil and clothes handed out to some pregnant women at the initial stage of the programme implementation but was stopped due to lack of fund to continue. |
| Facilities offering free healthcare | Pregnant women could access free maternal health care in all levels of government-owned hospitals, including the primary, secondary and tertiary facilities. | Pregnant women can only access free maternal healthcare at government-owned primary health care facilities, and delivery complications and caesarean section are managed for free, once referred from the PHC facilities. | Due to paucity of funds, pregnant women could only access free maternal health care in selected government-owned hospitals, which is not limited to primary, secondary or tertiary facilities. However, the distribution of health facilities is favourable to urban and peri-urban areas. |
| Funding | The state government mainly funded the programme and was supported by the federal government under the subsidy reinvestment programme as well as many international donor agencies and partners. | Funded by the state government and by the federal government under the subsidy reinvestment programme | Funded by the state government and by the federal government under the subsidy reinvestment programme |
| Health system strengthening to manage the anticipated influx of pregnant women | Two new ultra-modern tertiary health facilities were built, and many primary health care facilities were refurbished as well as new primary facilities built at settings where there are no facilities. Besides these, more health workers, including community health workers, health assistants, doctors, and nurses were recruited in large numbers | Provision of potable water to facilities lacking access to water. Renovation of some primary health facilities in bad shape. Recruitment of support staff on a temporary contract. | There was the recruitment of support staff on a temporary contract. |

participants. The distribution of participants per interviews and focus group discussions is presented in Table 2.

A minimum of six women took part in each focus group discussion, and each group contained women of similar demographic characteristics (18–25 years, 26–34 years and 35 years and above). Each session started with the introduction of the purpose of the study and completion of consent forms. The author and a graduate student moderated all the discussions and were assisted by trained research assistants tasked with note-taking and handling of audio recordings which allowed the researcher to maintain eye contact with the participants. The researcher ensured that each participant contributed and no particular participant was allowed to dominate the discussions. The researcher maintained open-mindedness and skills in eliciting information, the climate was non-threatening, and all the participants were introduced to

**Table 2. Participants selection in the qualitative study.**

|  | IDI | FGD |
|---|---|---|
| Total participants | 68 | 42 |
| State |  |  |
| Ekiti | 22 | 14 |
| Ondo | 23 | 16 |
| Nasarawa | 23 | 12 |
| Age |  |  |
| 18–25 | 21 | 16 |
| 26–34 | 23 | 14 |
| 35 and above | 24 | 12 |

one another. The participants sat in a circle for better communication to ensure productivity as well as comfort in disclosing information. The IDI lasted for an average of 50 minutes, while the FGDs lasted for an average of 110 minutes. The author developed an interview guide containing prompts and guides, which was assessed by experts in qualitative studies and later piloted among women who were not included in the main study. The interview guide was based on an extensive review of the literature. Data saturation was achieved after conducting over 22 interviews in each state.

All focus group discussions and interviews were tape-recorded, and notes were taken to complement the audio recordings. The interviews and focus group discussions were conducted in the local languages, and the transcripts were translated into English. Backward translation was performed by a professional translator to ensure the accuracy of translations. All participants had at the outset granted permission for the interviews to be audio recorded.

The author, a PhD student, together with 16 research assistants, made up of graduate students and final year undergraduate students, conducted the survey. The research assistants were trained on administering the questionnaire and on the ethical principles guiding the study. Structured questionnaires were administered to 1227 women, aged 18 years and above that had given birth in the previous five years preceding the survey (2011–2015). The selection criteria for the 1227 women follows: since no official data were available on the number of women that had given birth in the previous five years preceding the survey, the formula for calculating the sample size for 'infinite' population was employed[43]. At a confidence level of 95%, the precision level of +/-5%, and with an 'infinite' population, the appropriate sample size per state was 382. However, due to anticipated non-completion and missing responses, an additional 10% of the sample size was added. Thus, a total of 1227 respondents participated in the study. Participants were selected using a three-stage cluster random sampling. Each state was clustered into enumeration units and was stratified based on rural areas, towns, and cities. Simple random sampling was employed to select Enumeration Areas (EAs) from the updated list of EAs in the Nigerian 2006 census, with probability proportional to size. Approximately 25 clusters per state were required to achieve the sample size, and 15 to 30 households were randomly selected in each EA. In each EA, every tenth household was selected, and we skipped households where no woman met the selection criteria until the sample size was reached. However, the analysis in this paper was limited to 407 mothers who self-reported that they had benefitted from the user fee removal policy and did not pay any fee related to child delivery.

Before proceeding with this study, ethical approval was sought from the University of Fort Hare and Ondo State Ministry of Health Ethical Review Committee. This was granted. Every participant was taken through the aims of the study, and use of de-identified data before interviews were held; they were also educated on their right to withdraw from the sessions at any

time if they felt uncomfortable. A consent form was handed to each participant to indicate their willingness to be part of the study and, where appropriate, verbal consents were obtained.

## Variables and measurements

The questionnaire for the larger study was developed based on a review of literature [14–16, 19, 20] and extraction of relevant questions from Nigeria Demographic and Health Survey [44]. The instrument was assessed for face and construct valid by peers and project supervisor. Also, the instrument was piloted among mothers in another Nigerian state not included in the study and feedback on the clarity of the questions were implemented.

Satisfaction with care was assessed by focusing on the three central dimensions identified by Bramadat et al. [23]. The questions covered the structural aspect of care, such as the cleanliness of wards, theatres and toilets; waiting time; privacy; and adequate human and material resources. Beneficiaries of free maternal health care were asked to rate the cleanliness of the wards, theatres and toilets from very clean to very dirty. Also, they were asked to rate the waiting time from very short to very long. Privacy was measured by asking beneficiaries of free maternal health care whether their privacy was ensured with binary responses of "yes or no" provided for participants to choose from. Also, respondents were asked to state whether medicines required were available or not.

Women were also asked to assess the interpersonal aspect of care by stating their level of satisfaction with the attitude of the nurses and doctors. The responses provided include very good, good, average, bad and very bad. In terms of outcome dimensions, participants were asked to describe the aspect of care they were most satisfied with and their overall level of satisfaction with the services.

The sociodemographic characteristics of study participants, including age, parity, level of education, income, ownership of bank accounts, television, mobile phone and access to the internet, were obtained. Age was measured as a continuous variable but later grouped. Mothers were asked to state the number of children ever born and those born between 2011 and 2015. Level of education was assessed by asking respondents to state the highest level of education attained, and responses were group under no formal education, primary education, secondary education and tertiary education. Ownership of mobile phone, bank account, television and access to the internet were measured as categorical variables with a binary response of "yes or no".

## Data analysis

Data were coded and captured with the aid of the Statistical Package for Social Sciences (SPSS, version 24). A descriptive statistic involving frequency distribution was performed for all variables of interest. Qualitative data were analysed using thematic analysis. Data were transcribed and read carefully as a familiarisation technique. This was followed by a more in-depth reading during which codes were assigned to responses. The emerging codes were grouped under the most appropriate themes. The notes taken during the in-depth interviews and focus groups were revisited to ensure data had not been lost to translation and that interpretation had not been misleading. An expert in qualitative research, who is a colleague, was asked to peer review the emerging codes and themes so as to validate the analysis.

## Results

The analysis was performed on 407 women who had benefitted from user fee removal for maternal health care services, including antenatal-, delivery- and postnatal care. Over half of these women were from Ondo state where user fee removal had been implemented in all

government-owned health facilities, unlike Ekiti state where free maternal health care services had only been accessible at primary health care facilities (Table 3). Most participants were Christian (78.9%), currently married (95.8%), owned a mobile phone (94.6%), watched

Table 3. Socio-demographic characteristics of respondents.

| Variable | Frequency | Percentage |
|---|---|---|
| **State** | | |
| Ekiti | 116 | 28.5 |
| Ondo | 209 | 51.4 |
| Nasarawa | 82 | 20.1 |
| **Place of residence** | | |
| City | 138 | 33.9 |
| Town | 115 | 28.3 |
| Rural | 154 | 37.8 |
| **Religion** | | |
| Christian | 321 | 78.9 |
| Islamic | 85 | 20.9 |
| Traditional | 1 | 0.2 |
| **Level of education** | | |
| No formal education | 14 | 3.4 |
| Primary education | 64 | 15.7 |
| Secondary education | 211 | 51.8 |
| Higher education | 118 | 29.0 |
| **Marital status** | | |
| Currently married | 390 | 95.8 |
| Previously married | 2 | 0.5 |
| Never married | 15 | 3.7 |
| **Own a mobile phone** | | |
| Yes | 385 | 94.6 |
| No | 22 | 5.4 |
| **Regularly watch television** | | |
| Yes | 379 | 93.1 |
| No | 28 | 6.9 |
| **Own a bank account** | | |
| Yes | 213 | 52.3 |
| No | 194 | 47.7 |
| **Have access to the internet** | | |
| Yes | 122 | 30.0 |
| No | 285 | 70.0 |
| **Employed** | | |
| Yes | 323 | 79.4 |
| No | 84 | 20.6 |
| **Parity** | | |
| 1 | 89 | 21.9 |
| 2 | 103 | 25.3 |
| 3 | 95 | 23.3 |
| 4 | 80 | 19.7 |
| 5–8 children | 40 | 9.8 |
| **Income per month** | | |
| 50 dollars and above | 79 | 19.4 |
| Less than 50 dollars | 328 | 80.6 |

television regularly (93.1%), were employed (79.4%) but earned 50 dollars or less monthly (80.6%).

## Level of satisfaction with maternal health care services

As shown in Table 4, a vast majority of the participants were satisfied with all the aspects of care, including structural, process and outcome dimensions. Almost all participants (94.3%) described the facilities as clean or very clean. A vast majority affirmed that medicine and delivery kits were available and dispensed freely. On the process dimension of care, a vast majority of women described the nurses and doctors' attitudes as good or very good. Likewise, almost all participants (99.0%) stated that their privacy had been ensured.

In terms of the outcome dimension of care, 97.1% of beneficiaries were satisfied overall with the care received under free maternal health policy. For most of the users of the free maternal health care services that reported satisfaction, their successful birth outcomes, that is, having a successful delivery with a healthy baby and good personal health, were the most critical reason given for their satisfaction aside from user fees exemption, while an adverse outcome, such as death of their baby, injury and complications attributable to health workers negligence, was the main reason for dissatisfaction.

## Reasons for satisfaction with free maternal health care services

The qualitative study also supports the quantitative findings. Almost all the women indicated that they were satisfied with the services given and their successful birth outcomes, irrespective of their negative experiences during antenatal care and child delivery. The common response emerging from the interviews was, *"I am alive, and my baby is alive, I am satisfied"*. In all study areas, many beneficiaries of free maternal healthcare expressed their satisfaction with the programme and wanted the government to sustain the policy. To many women, free maternal healthcare is an excellent initiative, and the government deserves commendation for instituting such a policy. Specifically, most women were satisfied with the total removal of user fees because they did not have to worry about the cost of childbirth or antenatal services since those services were freely available. Women that delivered through caesarean section were particularly satisfied as many of them revealed that the cost of the caesarean section is unaffordable and they would not have been able to raise the money if there had been no free maternal healthcare. This view is supported by a middle-aged woman who registered with the primary health care centre to access a free caesarean section, given that only pregnant women who were registered with primary health centres qualified for free caesarean sections in Ekiti State. She had delivered her first child via caesarean section and had been told that subsequent babies might need to be through caesarean section also. She described her struggles to pay for the previous caesarean section and how that had motivated her decision to seek antenatal care in a primary healthcare centre so as to enable her to benefit from the user fee removal for caesarean section.

> *I benefited from the free health care programme, and I am satisfied with the care I received. As you may know, free health care is only accessible in primary health care facilities in Ekiti state, and only those who are referred for caesarean section from primary health care facilities are exempted from paying in tertiary facilities. I delivered my first child via caesarean section and I had been told that subsequent births will be through caesarean section, so, I had to register in a primary health care facility to benefit. The cost of caesarean section is too much and I knew we could not afford it for the birth of our second child. (Interview participant)*

**Table 4. Assessment of aspect of care and user satisfaction with services.**

| Variable | Frequency | percentage |
|---|---|---|
| **Structural aspect of care** | | |
| Assessment of cleanliness of wards and hospital | | |
| Very clean | 213 | 52.3 |
| Clean | 171 | 42.0 |
| Moderately clean | 17 | 4.2 |
| Dirty | 4 | 1.0 |
| Very dirty | 2 | 0.5 |
| Assessment of toilet conditions | | |
| Very clean | 170 | 41.8 |
| Clean | 203 | 49.9 |
| Moderately clean | 23 | 5.7 |
| Dirty | 7 | 1.7 |
| Very dirty | 4 | 1.0 |
| All medicine needed available | | |
| Yes | 378 | 92.9 |
| No | 29 | 7.1 |
| Ease of access to medicine | | |
| Very easy | 286 | 70.3 |
| Easy | 102 | 25.1 |
| Moderately easy | 14 | 3.4 |
| Not easy | 5 | 1.2 |
| Received drugs and delivery kits for free | | |
| Yes | 303 | 74.4 |
| No | 104 | 25.6 |
| Was the delivery 100% free? | | |
| Yes | 348 | 85.5 |
| No | 59 | 14.5 |
| **Process dimension of satisfaction** | | |
| Assessment of nurses' attitudes | | |
| Very good | 217 | 53.3 |
| Good | 167 | 41.0 |
| Average | 16 | 3.9 |
| Bad | 5 | 1.2 |
| Very bad | 2 | 0.5 |
| Assessment of doctors' attitudes | | |
| Very good | 205 | 50.4 |
| Good | 194 | 47.7 |
| Average | 7 | 1.7 |
| Bad | 1 | 0.2 |
| Was privacy guaranteed? | | |
| Yes | 403 | 99.0 |
| No | 4 | 1.0 |
| Were you attended to early at delivery? | | |
| Yes | 395 | 97.1 |
| No | 12 | 2.9 |
| **Outcome dimension of care** | | |
| Satisfied with quality of services received | | |

*(Continued)*

**Table 4.** (Continued)

| Variable | Frequency | percentage |
|---|---|---|
| Yes | 395 | 97.1 |
| No | 12 | 2.9 |
| Reasons for satisfaction | | |
| Good health of mother and baby | 379 | 95.9 |
| Fee removal, nurses attitude and "good Mama" package | 16 | 4.1 |
| Reasons for dissatisfaction | | |
| Adverse outcomes | 10 | 83.3 |
| Attitude of health workers, inadequate resources, waiting time | 2 | 16.7 |

Also, some women linked their satisfaction not only to the total removal of fees but also to items distributed to them during childbirth. Delivery kits containing baby clothes, soaps, Dettol, baby supplies were handed out to women after giving birth; many women were particularly happy with the government because of this initiative. For instance, in Ondo state, a 35 years old mother of two children who received a mobile phone was delighted about the initiative and commented thus:

> I enjoyed the service provided under free maternal healthcare. I am 100% satisfied; I was even given a mobile phone to contact doctors in case of an emergency. I am so grateful to the government for introducing this programme (In-depth interview participant 35, Ondo state).

This participant provided a mobile phone took part in the piloting of the safe motherhood programme in Ondo state when the feasibility of mobile phone intervention was tested. The scale-up of the programme, however, did not involve mobile phone intervention. Another woman, a 28-year-old mother of two children also from Ondo state, shared:

> I am satisfied because I was treated well and given a delivery kit. A white nurse took the delivery of my first child and provided good quality care (In-depth interview participant 36, Ondo state).

It is, however, important to note that delivery kits were given to pregnant mothers at the initial stage of the programme initiation and a majority of women in this study did not benefit from this gesture.

Also, most women expressed satisfaction with the cleanliness of the facilities, health workers' attitude, privacy and availability of resources. In Ondo state, in particular, several women were thrilled with the extent of infrastructure, equipment and doctors available in the newly built 'Mother and Child' facilities. Some women travelled far from their homes and even bypassed many health facilities to benefit from the perceived good quality of care delivered in the newly built facilities.

## Reasons for dissatisfaction

Despite the high level of satisfaction recorded, the results of the qualitative study revealed that a few women across the study area experienced prolonged waiting-time during antenatal care, verbal abuse by health workers, experience of adverse outcomes such as loss of infants and developing complications and some fee charges and difficulty in accessing drugs during antenatal care and childbirth. Their comments highlight the gaps in the quality of care rendered and could serve as pointers to where improvements are urgently required. Based on the

experiences of beneficiaries of free maternal healthcare, the reasons for dissatisfaction under free healthcare are discussed under specific subthemes below:

Adverse outcomes

Consistent with the survey findings, the experience of adverse outcome was among the main reason for dissatisfaction with care under the free maternal health care programme. The experience of one middle-aged participant underscores this point:

*I used free maternal healthcare for the birth of my third child and the baby died due to the negligence of the nurses. It is better to pay than experience that again. The things they are supposed to tell you will be hidden from you because they think you cannot pay. The medicine they are supposed to give will not be given because you are not going to pay. Also, they will not inform you of some medications you need because they have concluded in their mind that you do not have money (In-depth interview participant 8, Ekiti State, 13 July 2016).*

Another sad story emanating from the interview was the case of a mother who had previously given birth to five girls and was desperate for a male child. She visited the clinic for delivery and was told that she should wait that it was not yet time. In her words:

"*The nurses left me to watch a movie, and I delivered on my own, and the baby fell off and hit his head on the floor and died. It was painful for us as a family given that we were desperate for a male child.* I am not sure I can ever forgive the nurses for their negligence (Interview participant 6, Ekiti State).*

**Prolonged waiting-time.** Some women, mainly those residing in urban areas, were discontent with prolonged waiting-time during antenatal care in facilities offering free maternal healthcare. Of course, two reasons may be advanced for this, namely: insufficient doctors and overcrowding effect of free maternal healthcare. Nonetheless, some women believed that health workers were not steadfast enough in doing their jobs. A middle-aged mother of two children in Ondo state mentioned this during the in-depth interview:

*Waiting-time is too long, especially if you need to see a doctor. It is better to see the same doctor in his private practice and pay than to wait forever in the queue. As a businesswoman, I cannot wait forever; the doctor comes late to work because she would attend to patients in her private practice before coming to work (at the primary healthcare facility). Likewise, nurses in the antenatal clinic do not speak English during antenatal classes, so I do not get to understand the information passed on (In-depth interview participant 41, Ondo state).*

She is from the Igbo ethnic group, a minority group in Ondo State, Nigeria, where the Yoruba language is the popular language. Even though the number of doctors in government hospitals may be inadequate, patients did not necessarily acknowledge that and believed that the doctors were lazy at their jobs. In the study setting, women often woke up very early in the morning to join a queue of patients waiting to see a doctor, but some returned home without having accessed the services of the doctors they had sought. In the in-depth interview, a 24-year-old woman specified the time at which she usually rose to see a doctor:

*I wake up around 5 am anytime I want to see a doctor during antenatal care because if I do not, the queue will be too long and I may end up being unable to see the doctor that day (In-depth interview participant 35, Ondo state).*

Another woman described her experience of prolonged waiting-time in the FGD:

*I received antenatal and delivery care under free maternal healthcare, but I was unimpressed with the quality of care I received. The waiting time was too long; of course, I am aware that the patronage rose due to quality equipment and qualified health workers in the centre. Specifically, we were about 100 in the queue, and after attending to 20 of us, they stopped attending to us. We had to beg and beg because having waited for more than 6 hours without seeing the doctor would have amounted to a waste of the whole day. Because it was free, one could not complain. When I become pregnant again, I will not use the facility because they did not treat me well, even during childbirth. Visitors were not allowed to come and see me in the ward, but I still thank God because I delivered successfully (FGD participant 14, Ondo state).*

Beside the prolong waiting time, favouritism in managing the queue is another reason for user dissatisfaction. A 32- year-old woman with two children alleged that some women that did not come early like others and were seen before those in the queue, she even suggested that if one did not know someone at the hospital, then one would not receive care. She stated:

*There was a prolonged waiting-time during antenatal care. The health workers were biased in attending to pregnant women; if you did not know a nurse there, you would not receive care (In-depth interview participant 35, Ondo state).*

**Mistreatment, abuse and disrespect.** Another important reason why women were dissatisfied with the quality of care received under free maternal healthcare was their mistreatment, abuse and disrespect. Respectful treatment entails swift action, empathy and use of professional language in addressing patients[45, 46]. Of course not all women interviewed had specific experiences related to poor treatment by health workers; however, there were enough grounds to establish that health workers, most notably the nurses, still considered themselves to be superior to the patients and as such, talked down to patients in the study area. A 35-year-old mother of two children who had no formal education complained about the abuse she suffered from a nurse in Ondo state:

*Initially, I sought care under free healthcare in [the] government hospital, but I was not attended to properly. I was abused several times; the doctors and nurses are bad people. I opted for the private hospital for childbirth because I could not endure it anymore (In-depth interview participant 35, Ondo state).*

A more compelling case was the experience of a 28-year-old mother of two children. She recounted her experience during the FGD:

*My experience during the birth of my first child nearly made me deliver my second baby in my shop. I did not even attend antenatal care in this clinic (name withheld) because I was scared of the nurses; I only registered at the clinic about a week before the delivery of my second child. During the birth of my first child, I was so weak and was too tired to walk about as the nurses instructed. So, I took time to rest, and I slept off at the back of the hospital. When the nurses saw me sleeping, they screamed at me and talked down to me. I was embarrassed, but it did not stop there, as I made my way to the ward, water and blood came out of me, and I was told to clean it up. I passed out while trying to clean the water and blood and was rushed to the theatre. I thought I was going to die. I only went there to deliver my second child because of lack of money, but my experience during the birth of my second child was excellent, and I*

*have friends with the nurses at the clinic now and I am sure the nurses will treat me preferentially going forward (FGD participant 35, Ondo state).*

The striking thing about her experience is that the nurses were trying to look after her own interests, but the tone of their message was unfriendly. Clearly, patients-health workers' communication needs improving. Another 25-year-old woman shared her experience:

*I do not like the way they treat pregnant women. The way they talk down to us is not right. You will ask questions, and they will not answer and look at your face as if you are not a human being. Women are in pain; hence, they deserve some respect and better treatment. But I must say that not all the nurses are bad (In-depth interview participant 31, Ondo state).*

A 27-year-old woman was put off by the attitude of health workers and decided to seek care in a private hospital. She recounted her experience in the in-depth interview:

*During the birth of my first child, I sought care under the free healthcare, but shockingly I was told there was no bed space and that I should come back. But we did not return because of what we observed. Pregnant women in labour were shouted at; they were told if they delivered on the floor, they would pack the baby themselves and also clean the floor. So we had the caesarean section for the birth of my first child in the private hospital. However, due to lack of money, I had to endure the attitude of nurses during the birth of my second child. The hospital has adequate equipment and that is the reason I like to deliver there. I believe if they charge fees and we have to pay, they will not address pregnant women in that manner. It is my money, and I would not take the nurse's attitude (In-depth interview participant 27, Ondo state).*

Aside from how the nurses talk down to patients, their disposition after informing pregnant women that it is not yet time for child delivery during labour is another unfavourable attitude of nurses that women complained about. Most women believed that nurses did not care enough about their pains nor show empathy towards them. Some mentioned that after telling them that they were not ready for delivery, nurses went back to chat or watch movies, and some women ended up delivering their babies all alone without the nurses' assistance. To them, this amounted to the negligence of duty and could result in harm to both the mother and the baby. A 28-year-old mother of three children elucidated upon this view in the in-depth interview:

*"Oga ta oga o ta owo alaru ape" (Workers will earn their wages whether the boss makes enough money or not), health workers in the government hospitals do not take their job seriously. My husband regretted taking me to a government hospital. We could not afford the user fees in a private hospital because my husband had lost his job. I am dissatisfied with the way the nurses attended to me. I told them I was ready to deliver, but I was told I was not yet ready. I delivered on my own while they were busy watching movies; my baby could have fallen off the couch and died if not for God (In-depth interview participant 31, Ondo state).*

**The limited scope of cover of free healthcare**

Aside from prolonged waiting-time and mistreatment, disrespect and abuse, the limited scope of coverage under free healthcare was another reason some women being dissatisfied with the quality of service received. Many women came to the hospital, expecting everything to be provided for free; however, their expectations were dashed when they were told to pay for delivery items. A middle-aged mother of two who experienced this even concluded that

there is no free maternal healthcare in Ondo state because she was made to pay for delivery items and registration fees. She shared the following in the in-depth interview:

> I was disappointed when I got there, and I was told to buy delivery items, which I supposed should be free. I ended up spending 10,000 Naira. I spent 7,000 Naira on delivery items alone and 1000 for registration and 2000 naira on medicines. There is no free maternal health in Ondo state; it is mere propaganda (In-depth interview participant 41, Ondo state).

Another woman who acknowledged that caesarian section was done for free, argued that women still have to pay for medicines and delivery items. She explained the reason why she was dissatisfied, below:

> Although the caesarian section was done for me for free, drugs were not free; my husband donated blood and paid for delivery items. I do not want my husband to donate blood. That is the reason I am not satisfied with the service (FGD participant 14, Ondo state, ).

A 28-year-old woman who previously utilised private clinics but switched to enjoy free maternal healthcare was disappointed with the scope of the coverage. She compared the scope of maternal healthcare in private hospital to that of free maternal healthcare, saying:

> There is limited coverage under free maternal healthcare. It does not cover special cases and lab tests. The care is not comparable to what one enjoys in a private hospital (In-depth Interview participant 14, Ekiti state).

Contrastingly, some women were dissatisfied not because of having to buy delivery items, but due to the rejection of what they bought before coming to the health centre. A 32-year-old mother of two children was livid about her experience:

> I was told to buy plastic for disposal of the umbilical cord. Tell your wife to give birth in a private hospital. It is better to pay and be taken good care of. If you do not pay 2000 naira, they will not attend to you, even if you are due for delivery. The nurses are like Margaret Thatcher. I bought some delivery items and was told to buy the same items in the hospital without explaining to me why they had rejected the ones I had brought. Specifically, I was made to pay for another glove. The waiting time is too long, and if your delivery items are incomplete, you will not be attended to (In-depth interview participant 35, Ondo state).

It is important to emphasise that this particular participant did not register to obtain a safe delivery card, which is mandatory. The 2000 Naira referenced is a penalty for late registration rather than informal payment or bribe.

**Inadequate equipment and birth space.** In urban parts of Ondo state, there were complaints about inadequate postnatal care wards. Of course, this could not be divorced from the overcrowding effect of free maternal healthcare. Nonetheless, this cannot be overlooked because it impacted on the rating of the quality of care under free maternal healthcare. A 28-year-old woman summarised the shortage of equipment and birth space in the IDI as follows:

> The beds were insufficient. When new patients came, they asked older patients to either sleep on the floor or go home. I was told to buy all delivery items there in the hospital, and those I

*brought were rejected. There was no ultrasound machine, unlike the private hospital where I started antenatal care (In-depth interview participant 36, Ondo state).*

Another woman, who had to deliver by herself while waiting to get a space in the theatre because there was no space for her, recounted her experience:

*There were too many people, and the available birth spaces were inadequate, so I delivered by myself before they could attend to me. There is nothing like a private hospital. Public hospitals are not without complaints. The care I received is not good enough (In-depth interview participant 24, Ondo state).*

Some women had to deliver babies on the floor due to lack of birth space. Also, due to inadequate postnatal care wards, women were discharged quickly. Women that delivered through caesarean section were discharged, on average, three days after surgery. Some women, that were used to three to seven days of postnatal care depending on the mode of delivery, were particularly concerned about the limited time for postnatal care under free maternal healthcare.

**Bureaucratic inefficiency.** Lastly, bureaucratic inefficiency was reported as one of the reasons why women were dissatisfied with the quality of care under free maternal healthcare. To benefit from free maternal healthcare, pregnant women pass through a stressful registration process to obtain the registration card, called "*Ibi Ayo* card" in Ondo state–a term that means 'successful birth'. For some reasons, some women did not register before child delivery time and were made to pay triple the amount for registration during childbirth. Since registration must be done before giving care, a few women in labour pain ended up giving birth on their own at the entrance of the hospital because they would not admit them to the hospital without the card. A 35-year-old mother of four children recounted this during the FGD:

*The attitude of the nurses is poor. If you do not have the "ibi ayo" card, they will not attend to you. They took money from my friend (6000 Naira) and still asked her to buy some drugs. If you do not buy drugs and other delivery items, they will not attend to you. I saw a woman that gave birth at the entrance of the hospital because the nurses insisted on the registration card. I do not think that is good enough; they should be more humane (FGD participant 22, Ondo state).*

Also, the registration process was reported to be cumbersome in Ekiti due to the complicated processes and prolonged waiting-time; this probably resulted from the many women registering at the same time. A woman described the processes of card registration in Ekiti State during the in-depth interview:

*The process of registering at the health centre offering free healthcare was frustrating. One would pay at the bank and then return with the teller to their headquarters. The queue during this process was too long, and before it got to your turn, it was very difficult (In-depth interview participant 23, Ekiti state).*

Also, in rare cases, health workers did not perform a thorough check-up because there were too many patients. Throughout the interviews, only one woman alleged this. She is a middle-aged woman who stated:

*Because there are too many people, the health workers did not do a proper checkup. They were always in a hurry, but although one is really sick during pregnancy, they did not take much time to examine you (In-depth interview participant 46, Nasarawa state).*

## Discussion

This study was undertaken to examine the level of user satisfaction, reasons for satisfaction and dissatisfaction in the context of user fee removal policy. The study found that the vast majority of women were satisfied with the care received, irrespective of their complaints or criticism of the services. Existing studies on satisfaction with maternal health care services have consistently reported a high level of user satisfaction, which is consistent with our results [25–29, 47]. This finding is encouraging and suggests that health facilities will retain beneficiaries of free maternal health care.

The main reason for the high level of maternal satisfaction is a successful birth outcome, which almost all participants enjoyed. High overall satisfaction is important; however, this finding must be put into context. In an environment where user expectations are low, and most facilities in the country lack the necessary infrastructure and human resources to function optimally [31–33, 35, 48], having a successful birth—healthy mother and healthy baby–is not always guaranteed and, when achieved, is considered an enormous achievement. As such, it is not surprising that having a successful birth was the main reason most women put forward as the overarching factor determining their satisfaction. It is also possible that women sincerely appreciate the efforts of health workers who work under impossible conditions, with a heavy workload, limited resources, and little rewards and yet still manage to deliver positive outcomes for them and their babies.

Another plausible reason for this could be that the study participants are only a subset of all women who gave birth over the period. In other words, women who viewed facilities offering free maternal healthcare services as being those that provide poor quality service, may have sought services elsewhere. Research has shown that the perceived quality of care influences the choice of places to seek maternal health care services [49]. Given that most of our beneficiaries of free maternal healthcare services in this study earned below 100 dollars per month, it is plausible that they lacked real alternative services to utilise. Also, for women residing in rural areas, the chances are that they only have access to one facility. The lack of real alternative means there is no basis for comparison; thus, the expectation of quality services is very low.

Also, the removal of user fee appears to influence the level of satisfaction reported in this study, with several women indicating they were pleased with the removal of the user fee and with the delivery kit handed out to them. The study reveals that those beneficiaries were pleased, not only because of their successful birth outcomes, but also due to the cleanliness of the health facilities, privacy, attitude of health workers and provision of "*good mama package*"(delivery kit) and the user fee removal. It is, however, important to note that delivery kits were given to pregnant mothers at the initial stage of the programme initiation and a majority of women in this study did not benefit from this gesture. A similar study in South Africa—where free maternal health care was implemented during the 1990s— shows that mothers were mostly satisfied with the general cleanliness of the ward; the information provided by nurses about looking after themselves and their babies at home, including breastfeeding; the way privacy was maintained; and the thoroughness of examinations done by doctors and midwives [27].

Despite the high level of satisfaction reported in this study, a few women complained about structural and process dimensions of care and blamed the removal of user fee as the reason for

their poor treatment. From the qualitative analysis, prolonged waiting-time, the limited scope of coverage, disrespectful and abusive treatment, inadequate infrastructure and bed space and bureaucratic inefficiencies were the main reasons why a few women were dissatisfied with care given under free maternal health care, which is consistent with an existing study on the topic in Nigeria [50]. Previous studies have shown that provider's attitude and communication, as well as longer duration of stay in the ward, were independent predictors of client satisfaction [28]. Also, some studies have reported that mothers are disrespected and abused during child delivery in many settings in sub-Saharan Africa. A study characterised maternal health services in Nigeria as being of poor quality with pervasive abuse and lack of commitment to the needs of mothers and sensitivity to women [31]. Mistreatment and abuse of women seeking maternal health care services have been documented in all geopolitical zones of Nigeria [30, 34, 51, 52].

## Policy implications and recommendations

Despite the high level of user satisfaction in this study, there is a need to address the identified reasons for women's dissatisfaction with the quality of care under the fee removal regime. Specifically, health workers and mothers' communication needs to improve considerably, and waiting-time in urban facilities must be addressed, perhaps through establishing more health facilities. Also, the penalty for late registration for safe delivery card undermines efforts towards ensuring increased use of maternal health care services and thus, need to be removed. Likewise, the practice of mandating mothers, who had already purchased delivery items before coming health facilities, to repurchase those items mentioned by a few participants should be further investigated and discouraged. Addressing these issues will further inspire confidence in women who did not access care under the user fee removal policy due to a lack of trust in the quality of services rendered. Also, considering the high level of satisfaction among users, efforts should be concentrated not only towards the sustainability of the programme but also expanding the scope of coverage of services covered to include other services mothers want like ultrasound service and treatment of complications other than those requiring caesarean section. Free maternal health programme has been demonstrated to have a transformative effect including improving women's capability to make health decisions and their social position[53].

## Study strength and limitations

This study is not without some limitations. The views presented in this paper account for only users of free maternal health care, which may not have concerns about the quality of services rendered in the facilities. It is possible that women that sought care in facilities charging user fees have concerns about the quality of care in those not charging user fees. As such, the high level of satisfaction is not an indication of quality maternal health care services. Nonetheless, assessing the level of beneficiary satisfaction with care using a population-based survey, which allows women to freely and independently evaluate the services without the influence of health workers, is a strength of this study. Also, the use of a mixed-method design is another strength of this study because it allows for the generation of more nuanced findings.

## Conclusion

This study established that women are highly satisfied with care under the user fee removal policy and that they desired that the Nigerian government sustain the policy. There is generally a high level of satisfaction with structural and process dimensions; however, bureaucratic inefficiencies, prolonged waiting-time, poor health workers' attitudes and inadequate infrastructure must be addressed to maximise the gains of user fee removal policy.

## Supporting information

**S1 Data. Maternal satisfaction dataset.**
(SAV)

## Acknowledgments

I express my profound gratitude to the Govan Mbeki Research and Development Centre (GMRDC)—the research office of the University of Fort Hare—for providing bursary and fee waiver in support of my doctoral degree. Also, my endless gratitude goes to my research assistants, Tunde Awopegba, Roselyn Alademomi, Yomi Ojo, Yusuf Ishaya Madallah, Maikeffi John, Pricilla Christopher and Temitope Ojo, my project supervisor—Professor Wilson Akpan —and my study participants. I am grateful to the African Population and Health Research Center for my Postdoctoral Research Scientist position during which I completed this paper.

## Author Contributions

**Conceptualization:** Anthony Idowu Ajayi.

**Data curation:** Anthony Idowu Ajayi.

**Formal analysis:** Anthony Idowu Ajayi.

**Investigation:** Anthony Idowu Ajayi.

**Methodology:** Anthony Idowu Ajayi.

**Resources:** Anthony Idowu Ajayi.

**Supervision:** Anthony Idowu Ajayi.

**Validation:** Anthony Idowu Ajayi.

**Writing – original draft:** Anthony Idowu Ajayi.

**Writing – review & editing:** Anthony Idowu Ajayi.

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
