## [Decision Letter · Decision Letter 0]

30 Oct 2019

PONE-D-19-21477

“I am alive; my baby is alive”: Understanding reasons for satisfaction and dissatisfaction with maternal health care services in the context of user fee removal policy in Nigeria

PLOS ONE

Dear Dr Ajayi,

Thank you for submitting your manuscript to PLOS ONE. After careful consideration, we feel that it has merit but does not fully meet PLOS ONE’s publication criteria as it currently stands. Therefore, we invite you to submit a revised version of the manuscript that addresses the points raised during the review process.

The reviewers considered your manuscript to have merit and have made some very helpful suggestions which will improve the quality of the manuscript. In particular, Reviewer 1 highlights that more information is required in the methods (mixed-methods approach, sampling) and that a more in-depth approach to the qualitative data is needed (e.g. comparing between sites). Reviewer 2 also suggests a more balanced presentation of the qualitative results is needed.

We would appreciate receiving your revised manuscript by Dec 14 2019 11:59PM. To enhance the reproducibility of your results, we recommend that if applicable you deposit your laboratory protocols in protocols.io, where a protocol can be assigned its own identifier (DOI) such that it can be cited independently in the future. For instructions see: http://journals.plos.org/plosone/s/submission-guidelines#loc-laboratory-protocols

We look forward to receiving your revised manuscript.

Kind regards,

Alexandra Sawyer

Academic Editor

PLOS ONE

Journal Requirements:

2. We note you have included a table to which you do not refer in the text of your manuscript. Please ensure that you refer to Table 2 in your text; if accepted, production will need this reference to link the reader to the Table.

3. Please include a title for table 3.

Reviewers' comments:

Reviewer's Responses to Questions

**Comments to the Author**

1. Is the manuscript technically sound, and do the data support the conclusions?

Reviewer #1: Yes

Reviewer #2: Yes

2. Has the statistical analysis been performed appropriately and rigorously? 

Reviewer #1: N/A

Reviewer #2: Yes

3. Have the authors made all data underlying the findings in their manuscript fully available?

Reviewer #1: Yes

Reviewer #2: Yes

4. Is the manuscript presented in an intelligible fashion and written in standard English?

Reviewer #1: Yes

Reviewer #2: Yes

5. Review Comments to the Author

Reviewer #1: This article is interesting because it takes place in the context of a policy of free care. While the knowledge presented is not entirely original, it adds evidence to the challenges of perceiving the quality of free care in Africa.

I understand that this is an article that has already been revised because the second file shows changes.

The part that is really necessary to work on is the presentation of the method. What is the type of mixed method design? The presentation should follow the standards proposed by the MMAT. The integration of QUAL/QUANT data is not explicit and therefore it is not clear how this is really a mixed method. The sampling of qualitative interviews is not at all clear. On the basis of which theoretical or conceptual approach were the interview guides produced? It is surprising that the empirical saturation of 22 is exactly the same in each region, especially since two regions 23 have been achieved. For quantitative sampling the method deserves references and it is surprising that there is no official number of pregnant women in Nigeria as is the case elsewhere in Africa. The questionnaire seems to be based on Bramadat dimensions but nothing is known about the variables used and the validity of the questionnaire, especially since there is no statistical analysis but only descriptive analysis. Why not propose regression analyses for example to understand the factors associated with your dependent variables?

However, it is not easy to understand whether the results presented are really related to free care, the usual practice of care or other interventions (telephone distribution, delivery kits, etc.). The authors should take advantage of their 3 different contexts to better compare and contrast their results in the absence of free comparison. It is understood that the implementation ("variation in the implementation") and the content of the interventions are different between the 3 regions but no details are presented on these differences and they are not used to contrast the answers. The results are presented as if the situation were the same everywhere for all health centres and all women. Is that really the case?

This is a large-scale data collection and it is surprising that the author states that no funding has been used and that the study is signed by only one author given the scope of the process.

See : https://doi.org/10.1016/j.socscimed.2017.11.045

Reviewer #2: PLOS One Review

Title:

1. The title is catchy but the phrase ““I am alive; my baby is alive” does not appear in any of the quotes.

Introduction

2. Line 94…Include which year user fee removal policy was launched and what services it covered e.g. antenatal, delivery, postnatal services etc.

Methods

3. Line 116-117: “The choice of the two geopolitical zones is an obvious one because of the uniqueness of their free maternal health programmes.”. Consider including what aspects make the free maternal health programmes unique in the two geopolitical zones to give a bit of context about the free maternity programme although this has been alluded to on lines 206-209.

4. Lines 134-135: “An average of six women took part in each focus group discussion and each group contained women of similar demographic characteristics”. Indicate the actual demographic chacteristics for example age etc

5. Rephrase lines 137 to 139 to avoid repetition.

6. Line 193: The process of data analysis sounds more of thematic rather than content analysis.

7. Include which year the data was collected in the methods section. I don’t think it’s necessary to include the date and year of data collection after each quote in the results section.

Results

8. Line 227: For clarity purposes, explain what adverse outcomes means in this context for example, is it death of the child, death of the mother etc

9. Line 228: This should be Table 3 not Table 2

10. Line 327: Provide reference for this sentence.

11. Line 350-351: Needs to be rephrased to avoid misunderstanding the referenced quote with regards to the nurses’ behaviour.

12. Line 390-427: Under the section on limited scope of cover for free healthcare, it would be good to clarify if the fees charged for maternal health services were actually informal payments since all services were supposed to be provided for free. For example, line 421-422 “…If you do not pay 2000 naira they will not attend to you, even if you are due for delivery…” could this be alluding to informal payments/bribes.

14. Line 430 and 282 “crowding effect” is a little unclear, do you mean “overcrowding”?

15. Almost all of the quotes are from Ondo State with only one quote each from Ekiti and Nasarwa State, they should be more balance such that qualitative findings from all states are well represented.

Discussion

16. Lines 536-541: This sounds like recommendations. Consider including a subtitle to signpost this, for example; “Policy Implications/Recommendations”. You can also expand on the policy implications already provided and add more based on the study findings e.g. implications of continued charging of user fees, possibility of existence of informal fees/bribes

6. PLOS authors have the option to publish the peer review history of their article (what does this mean?). If published, this will include your full peer review and any attached files.

Reviewer #1: Yes: Valéry Ridde

Reviewer #2: No

---

## [Author Response · Author response to Decision Letter 0]

15 Nov 2019

Dear Editor, 

I am pleased to submit a revised version of my manuscript having attended to all comments raised by the reviewers. Special thanks to the reviewers for the expert and contructive comments which have further help me improve on my manuscript. A step-by-step response to all comments raised by the reviews could be found below. I trust you will find my manuscript in order.

Best Regards 

Anthony Ajayi

Journal Requirements:

Response 

The manuscript has been formatted in line with journal style

2. We note you have included a table to which you do not refer in the text of your manuscript. Please ensure that you refer to Table 2 in your text; if accepted, production will need this reference to link the reader to the Table.

Response

All Tables have been referenced in the manuscript. 

3. Please include a title for table 3.

Response

I have included a title for table 3.

Reviewers' comments:

Reviewer's Responses to Questions

Comments to the Author

1. Is the manuscript technically sound, and do the data support the conclusions?

Reviewer #1: Yes

Reviewer #2: Yes

2. Has the statistical analysis been performed appropriately and rigorously?

Reviewer #1: N/A

Reviewer #2: Yes

3. Have the authors made all data underlying the findings in their manuscript fully available?

Reviewer #1: Yes

Reviewer #2: Yes

4. Is the manuscript presented in an intelligible fashion and written in standard English?

Reviewer #1: Yes

Reviewer #2: Yes

5. Review Comments to the Author

Reviewer #1: This article is interesting because it takes place in the context of a policy of free care. While the knowledge presented is not entirely original, it adds evidence to the challenges of perceiving the quality of free care in Africa.

Response

We thank the reviewer for the positive feedback.

I understand that this is an article that has already been revised because the second file shows changes.

Response

The article was previously revised based on the comments of the editor.

The part that is really necessary to work on is the presentation of the method. What is the type of mixed method design? The presentation should follow the standards proposed by the MMAT. The integration of QUAL/QUANT data is not explicit and therefore it is not clear how this is really a mixed method. 

Response

I am grateful to the editor for this important comment. I have now indicated the type of mixed methods design employed in the study, provided rationale for using a mixed-method design and cited the MMAT guideline for reporting a mixed-methods study. 

The sampling of qualitative interviews is not at all clear. On the basis of which theoretical or conceptual approach was the interview guides produced? It is surprising that the empirical saturation of 22 is exactly the same in each region, especially since two regions 23 have been achieved. 

Response

The interview guide was not based on the theory of saturation. The development of instrument was based on a review of literature, especially by expert in the field like Bruno Meessen, Valéry Ridde, Christine Lagarde, Sophie Witter and Bramadat, to name a few as well as the DHS questionnaire. The interview guide was assessed by experts in the field of qualitative research before piloting and administering. 

However, the principle of theoretical saturation guided the recruitment of participants for the qualitative interviews (see Glaser, B. G. & Strauss, A. L. (1967). The discovery of grounded theory: Strategies for qualitative research. Piscataway, New Jersey: Transaction). Our goal was to gather sufficient data to answer the study objectives. The interviews continued until no new information emerges in the first study setting. We leverage on our experience in the first study setting and given that the study focused on a number of issues, we kept an open mind to reach 22 interviews in the remaining two study settings with the expectation that new information could emerge. 

For quantitative sampling the method deserves references and it is surprising that there is no official number of pregnant women in Nigeria as is the case elsewhere in Africa. 

Response

Reference has now been added. Data availability remain a major in Nigeria. The last census was done 13 years ago even though the law mandates a census every 10 years. 

The questionnaire seems to be based on Bramadat dimensions but nothing is known about the variables used and the validity of the questionnaire, especially since there is no statistical analysis but only descriptive analysis. Why not propose regression analyses for example to understand the factors associated with your dependent variables?

Response

I have discussed the validity of the questionnaire in the methods. Also, I have provided detailed information on how the variables were measured. I did not perform a regression analysis for two reasons. First, over 97 percent of the women stated that they were statisfied and only 12 women were not satisfied, with no variation by background characteristics. I believe the qualitative study helps to understand reasons for satisfaction and dissatisfaction, which did not really come out in the quantitative findings. 

However, it is not easy to understand whether the results presented are really related to free care, the usual practice of care or other interventions (telephone distribution, delivery kits, etc.). The authors should take advantage of their 3 different contexts to better compare and contrast their results in the absence of free comparison. 

Response

The reviewer raised an important point. While the provisioning of free health care for pregnant women and children less than 5 years dominated the news, it emerged from the study that adequate preparation was made to deal with the anticipated influx of mothers. There was health system strengthening, in terms of recruitment of additional health workers, provisioning of drugs and materials as well as some renovation of a few health facilities. We have described this elsewhere (see Ajayi A, Akpan W. Maternal Outcomes in the Context of Free Maternal Healthcare Provisioning in North Central and South Western Nigeria. In Studies in the Sociology of Population 2019 (pp. 301-318). Springer, Cham.). The focus of this paper was to assess the users’ satisfaction. All participants included in this paper benefited from free maternal health care services. Some benefitted from the pilot programme when the feasibility of mobile phone intervention and delivery kit were piloted. However successful the pilot was, it was never scaled-up at full implementation, and aside from one or two women who mentioned receiving delivery kit or mobile phone, a vast majority of the women in the study did not receive these items. I have stated this in the manuscript. I have also added a Table to explain the context of the programme in the three settings.

It is understood that the implementation ("variation in the implementation") and the content of the interventions are different between the 3 regions but no details are presented on these differences and they are not used to contrast the answers. The results are presented as if the situation were the same everywhere for all health centres and all women. Is that really the case?

Response

I have presented more information regarding the variations in the implementation (See Table 1). The variations in implementation may not necessarily affect the results presented given that the differences in implementation affects mainly the coverage of the programme or put differently the proportion of beneficiaries by study settings. In Ondo State, the programme has a wider reach given that women could access free maternal health care in all government-owned health facilties, while in Ekiti state, it was implemented only at primary health care and only those who were referred to higher level of care for birth complications were attended to for free in tertiary facilities. In Nasarawa state, many rural settings lack access to health facilities as such; the beneficiaries are mostly from urban areas. The results, in terms of beneficiary satisfaction did not differ. 

This is a large-scale data collection and it is surprising that the author states that no funding has been used and that the study is signed by only one author given the scope of the process.

Response

I have acknowledged all the people that contributed to this work, including my 14 research assistants and project supervisor. Also, I acknowledged the university for providing me with bursary towards the completion of my thesis. This is one of the papers from my PhD developed entirely by me. In terms of criteria for authorship, I am the sole person that meets required criteria. 

See : https://doi.org/10.1016/j.socscimed.2017.11.045

Response

Many thanks to the reviewer for identifying this literature that I completely missed while drafting this manuscript. I have read and referenced the paper. 

Reviewer #2: PLOS One Review

Title:

1. The title is catchy but the phrase ““I am alive; my baby is alive” does not appear in any of the quotes.

 Response

I have added the quote by adding transcript containing this quote. 

Introduction

2. Line 94…Include which year user fee removal policy was launched and what services it covered e.g. antenatal, delivery, postnatal services etc.

Response

Corrected has been effected.

Methods

3. Line 116-117: “The choice of the two geopolitical zones is an obvious one because of the uniqueness of their free maternal health programmes.”. Consider including what aspects make the free maternal health programmes unique in the two geopolitical zones to give a bit of context about the free maternity programme although this has been alluded to on lines 206-209.

Response

 I have now used a Table to specify the difference in the programmes by states.

4. Lines 134-135: “An average of six women took part in each focus group discussion and each group contained women of similar demographic characteristics”. Indicate the actual demographic characteristics for example age etc

Response

Done 

5. Rephrase lines 137 to 139 to avoid repetition.

Response

Done

6. Line 193: The process of data analysis sounds more of thematic rather than content analysis.

Response

I have changed to thematic analysis

7. Include which year the data was collected in the methods section. I don’t think it’s necessary to include the date and year of data collection after each quote in the results section.

Response 

Done

Results

8. Line 227: For clarity purposes, explain what adverse outcomes means in this context for example, is it death of the child, death of the mother etc

Response

 I have clarified what adverse outcomes mean. It is not limited to the death of the child but perceived negligence in the management of complications. For example, some women delivered on their own even when in the the clinic, with the nurses telling them that they are still not ready to deliver. There is a particular case while a woman delivered in her own and the baby fell and died. The woman wanted a male child, she had given birth to five children and the baby that died happens to be a male child. The woman blamed the the nurses for negligence accusing them of been busy watch TV instead of attending to her. I have included this case in the manuscript. 

9. Line 228: This should be Table 3 not Table 2

Response

Corrected 

10. Line 327: Provide reference for this sentence.

Response

Done

11. Line 350-351: Needs to be rephrased to avoid misunderstanding the referenced quote with regards to the nurses’ behaviour.

Response

Done

12. Line 390-427: Under the section on limited scope of cover for free healthcare, it would be good to clarify if the fees charged for maternal health services were actually informal payments since all services were supposed to be provided for free. For example, line 421-422 “…If you do not pay 2000 naira they will not attend to you, even if you are due for delivery…” could this be alluding to informal payments/bribes.

Response

It is difficult to tag the payment as bribes given that delivery items were given to them in return. The frustration the mother had relates to rejection of the items they brought from home and been made to purchase the same item at the clinic. Bribe may not be the appropriate term to use. Because this emanated from a qualitative interview, it is difficult to assess how widespread the practice is. Also, some women alluded to paying for registration fee, which is less than 2 dollars. Failure to register during pregnancy means, that mothers have to pay 13 dollars if they choose to deliver at health facilities without the registration card for pregnant mothers which they ought to have obtained during pregnancy. 

I have now added this statement to avoid the confusion: “It is important to emphasise that this particular participant did not register to obtain a safe delivery card, which is mandatory. The 2000 Niara referenced is a penalty for late registration rather than informal payment or bribe.” 

14. Line 430 and 282 “crowding effect” is a little unclear, do you mean “overcrowding”?

Response 

Changed to overcrowding

15. Almost all of the quotes are from Ondo State with only one quote each from Ekiti and Nasarwa State, they should be more balance such that qualitative findings from all states are well represented.

Response

I have now added more quotes from the two states. 

Discussion

16. Lines 536-541: This sounds like recommendations. Consider including a subtitle to signpost this, for example; “Policy Implications/Recommendations”. You can also expand on the policy implications already provided and add more based on the study findings e.g. implications of continued charging of user fees, possibility of existence of informal fees/bribes

 Response

Done

---

## [Decision Letter · Decision Letter 1]

11 Dec 2019

“I am alive; my baby is alive”: Understanding reasons for satisfaction and dissatisfaction with maternal health care services in the context of user fee removal policy in Nigeria

PONE-D-19-21477R1

Dear Dr. Ajayi,

We are pleased to inform you that your manuscript has been judged scientifically suitable for publication and will be formally accepted for publication once it complies with all outstanding technical requirements.

With kind regards,

Alexandra Sawyer

Academic Editor

PLOS ONE

Additional Editor Comments (optional):

Reviewers' comments:

Reviewer's Responses to Questions

**Comments to the Author**

1. If the authors have adequately addressed your comments raised in a previous round of review and you feel that this manuscript is now acceptable for publication, you may indicate that here to bypass the “Comments to the Author” section, enter your conflict of interest statement in the “Confidential to Editor” section, and submit your "Accept" recommendation.

Reviewer #1: All comments have been addressed

Reviewer #2: All comments have been addressed

2. Is the manuscript technically sound, and do the data support the conclusions?

Reviewer #1: Partly

Reviewer #2: Yes

3. Has the statistical analysis been performed appropriately and rigorously? 

Reviewer #1: N/A

Reviewer #2: Yes

4. Have the authors made all data underlying the findings in their manuscript fully available?

Reviewer #1: Yes

Reviewer #2: Yes

5. Is the manuscript presented in an intelligible fashion and written in standard English?

Reviewer #1: Yes

Reviewer #2: Yes

6. Review Comments to the Author

Reviewer #1: (No Response)

Reviewer #2: I am okay with the revised manuscript. The author has addressed all the comments I had raised following my review of the paper.

7. PLOS authors have the option to publish the peer review history of their article (what does this mean?). If published, this will include your full peer review and any attached files.

Reviewer #1: No

Reviewer #2: No

---

## [Editor Report · Acceptance letter]

16 Dec 2019

PONE-D-19-21477R1 

“I am alive; my baby is alive”: Understanding reasons for satisfaction and dissatisfaction with maternal health care services in the context of user fee removal policy in Nigeria 

Dear Dr. Ajayi:

I am pleased to inform you that your manuscript has been deemed suitable for publication in PLOS ONE. Congratulations! Your manuscript is now with our production department. 

With kind regards,

on behalf of

Dr. Alexandra Sawyer 

Academic Editor

PLOS ONE